# Synthetic Tryptanthrin Derivatives Induce Cell Cycle Arrest and Apoptosis via Akt and MAPKs in Human Hepatocellular Carcinoma Cells

**DOI:** 10.3390/biomedicines9111527

**Published:** 2021-10-24

**Authors:** Jing-Yan Gao, Chih-Shiang Chang, Jin-Cherng Lien, Ting-Wei Chen, Jing-Lan Hu, Jing-Ru Weng

**Affiliations:** 1School of Pharmacy, China Medical University, Taichung 40402, Taiwan; u100003056@cmu.edu.tw (J.-Y.G.); chihshiang@mail.cmu.edu.tw (C.-S.C.); jclien@mail.cmu.edu.tw (J.-C.L.); 2Drug Development Center, China Medical University, Taichung 40402, Taiwan; wei31999@gmail.com; 3Department of Marine Biotechnology and Resources, National Sun Yat-sen University, Kaohsiung 80424, Taiwan; annavsbelle@yahoo.com.tw; 4Graduate Institute of Natural Products, Kaohsiung Medical University, Kaohsiung 80708, Taiwan

**Keywords:** tryptanthrin, hepatocellular carcinoma, apoptosis, cell cycle arrest, ROS

## Abstract

Trytanthrin, found in Ban-Lan-Gen, is a natural product containing an indoloquinazoline moiety and has been shown to possess anti-inflammatory and anti-viral activities. Chronic inflammation and hepatitis B are known to be associated with the progression of hepatocellular carcinoma (HCC). In this study, a series of tryptanthrin derivatives were synthesized to generate potent anti-tumor agents against HCC. This effort yielded two compounds, A1 and A6, that exhibited multi-fold higher cytotoxicity in HCC cells than the parent compound. Flow cytometric analysis demonstrated that A1 and A6 caused S-phase arrest and downregulated the expression of cyclin A1, B1, CDK2, and p-CDC2. In addition to inducing caspase-dependent apoptosis, A1 and A6 exhibited similar regulation of the phosphorylation or expression of multiple signaling targets, including Akt, NF-κB, and mitogen-activated protein kinases. The anti-tumor activities of A1 and A6 were also attributable to the generation of reactive oxygen species, accompanied by an increase in p-p53 levels. Therefore, A1 and A6 have potential clinical applications since they target diverse aspects of cancer cell growth in HCC.

## 1. Introduction

Hepatocellular carcinoma (HCC) is the fourth leading cause of cancer-related deaths globally, with over 40,000 new cases and 30,000 deaths reported in the United States [1,2]. Obesity, alcohol consumption, smoking, exposure to aflatoxin, and viral infection (hepatitis B and C) are the common risk factors for HCC [1]. In addition to prevention, the current treatment options for HCC include surgery, liver transplant, chemotherapy, radiotherapy, and embolization [3]. However, the 5-year relative survival rate is lower than 20% due to tumor recurrence, cirrhosis, and organ shortage [4]. Thus, there is an urgent need to develop a new therapeutic strategy for the treatment of HCC. 

For centuries, natural products have been used as enriched sources for the prevention and therapy of various diseases, including cancer [5]. For example, camptothecin, an alkaloid from *Camptotheca acuminate*, has been used to treat stomach cancer, bladder cancer, and leukemia for over 30 years [6]. Paclitaxel, originally isolated from *Taxus brevifolia*, alters microtubule polymerization in the cell cycle and is a commonly prescribed anti-cancer drug approved for the treatment of breast cancer, ovarian cancer, lung cancer, and oral cancer by the U.S. Food and Drug Administration [7]. Similarly, vinorelbine, originally developed from vinca alkaloid, was approved as the first line of treatment for advanced lung cancer in 1994 [8]. 

Ban-Lan-Gen, a traditional Chinese medicine (TCM), has been used to relieve inflammation, mumps, hepatitis, and seasonal influenza [9,10,11]. Xiao et al. reported that the extract of Ban-Lan-Gen inhibits the production of inflammatory cytokines, including nitric oxide, prostaglandin E2, and tumor necrosis factor-α in lipopolysaccharide-treated RAW264.7 cells [12]. Chronic inflammation is associated with an increased risk of developing cancer, accounting for 15–20% of cancer-related deaths [13]. Notably, Ban-Lan-Gen is a TCM that is associated with a significantly reduced risk of HCC in hepatitis B-infected patients [14]. Natural products such as alkaloids partly contribute to the biological activity of Ban-Lan-Gen [15]. Tryptanthrin, an alkaloid isolated from Ban-Lan-Gen, causes G1 cell cycle arrest, and was found to downregulate the expression of cyclin D2 in murine myeloid leukemia cells [16]. Multiple studies have shown that tryptanthrin suppresses tumor growth by modulating various targets, including p38, ERK, PIM1 kinase, and MDR1 [17,18,19,20]. Recent modifications of tryptanthrin have paved a way to obtain different targeted drugs that include amino-tryptanthrin derivatives such as topoisomerase II (Topo II), *N*-benzyl tryptanthrin derivatives such as indoleamine 2,3-dioxygenase (IDO), and tryptophan 2,3-dioxygenase (TDO) dual inhibitors (Figure 1A) [21,22]. In this study, we report the use of tryptanthrin as the lead compound to synthesize a series of tryptanthrin derivatives with improved anti-tumor activity. The pharmacological mechanism of tryptanthrin derivatives was also investigated in HCC cells.

## 2. Materials and Methods

### 2.1. Reagents, Chemicals, Antibodies

Isatoic anhydride and indoline-2,3-dione (Sigma-Aldrich, St. Louis, MO, USA) were used as the starting material to synthesize tryptanthrin and tryptanthrin derivatives (Figure 1B, Appendix A). The identity and purity of these compounds were identified by proton magnetic resonance spectrometry, EI-MS, and high-performance liquid chromatography (Appendix A). Antibodies against p-Akt^Ser473^, Akt, p-p38^180/182Thr/Tyr^, p38, p-ERK^202/204Thr/Tyr^, ERK, JNK, p-JNK^183/185Thr/Tyr^, caspase-8, NF-κB, p-NF-κB^536Ser^, IκBα, CDK2, CDC-2, p-p53^15Ser^, p53, cyclin A1, cyclin B1, and p-CDC2^15Tyr^ were purchased from Cell Signaling Technology (Beverly, MA, USA). Antibody against fibrillarin was purchased from Signalway Antibody (College Park, MD, USA). β–actin antibody and other chemicals were obtained from Sigma-Aldrich (St. Louis, MO, USA). All agents were dissolved in DMSO and added to cells at a final DMSO concentration (0.1%).

### 2.2. Cell Culture 

Human hepatocellular cell lines Hep3B and SK-Hep1 were gifts from Dr. Po-Chen Chu (China Medical University) and maintained in Dulbecco’s modified Eagle’s medium/Nutrient Mixture (DMEM) (Invitrogen, Carlsbad, CA, USA). Cells were supplemented with 10% fetal bovine serum (FBS; Gibco, Grand Island, NY, USA) in a humidified incubator containing 5% CO_2_ at 37 °C.

### 2.3. Cell Viability Analysis 

The cell viability of the compounds was determined by 3-(4,5-dimethylthiazol-2-yl)-2,5-diphenyltetrazolium bromide (MTT) assays [23]. Briefly, 0.5 mg/mL MTT (100 μL) was added to a 96-well plate per well and incubated for 4 h at 37 °C. After removing the medium, the reduced MTT dye was solubilized in DMSO (200 μL per well). A SPECTROstar Nano spectrophotometer (BMG LABTECH, Ortenberg, Germany) was used to measure the absorbance at 570 nm.

### 2.4. Flow Cytometry 

Cells (2 × 10^5^/3 mL) were treated with DMSO or drugs for 48 h, stained with propidium iodide (PI) or annexin V-FITC, and PI according to the vendor’s protocol. For cell cycle analysis, cells were washed with ice-cold phosphate-buffered saline (PBS) twice, fixed in 70% cold ethanol for 4 h at 4 °C, then analyzed by the multicycle software (ModFit_T3.0). For apoptosis evaluation, cells were analyzed by using a BD FACSAria flow cytometer (Becton Dickinson, Heidelberg, Germany) and BD FACSDiva 6.1.3 software (BD). Caspase-3 activation was assessed using a FITC rabbit anti-active caspase-3 kit according to the vendor’s protocol (BD Pharmingen, San Diego, CA, USA).

### 2.5. Reactive Oxygen Species (ROS) Generation 

Cells (2 × 10^5^/3 mL) were treated with DMSO or drugs, stained by 2′,7′-dichlorodihydrofluorescein diacetate (DCFH-DA) (5 μmol/L) for ROS determination, respectively [24]. Then, cells were analyzed by fluorescence intensity using flow cytometry (Becton Dickinson, Heidelberg, Germany) and BD FACSDiva 6.1.3 software (BD).

### 2.6. Western Blotting 

Proteins from lysed HCC cells were prepared on SDS polyacrylamide gels and transferred onto the nitrocellulose membrane [24]. Then, the membranes were incubated with primary antibodies overnight and then the secondary antibodies. The blots were detected by enhanced chemiluminescence.

### 2.7. Preparation of Nuclear Extracts

Cells were treated with DMSO or drugs for 48 h. The nuclear extracts were prepared using NE-PER^TM^ Nuclear and Cytoplasmic Extraction Reagents (Thermo Fisher Scientific, Waltham, MA, USA) according to the manufacturer’s instructions. Then, the nuclear extracts were analyzed by Western blotting.

### 2.8. Statistical Analysis 

All experiments were performed with at least three replicates, and results are represented as means ± standard deviation (SD) except indicated otherwise. All of the data were analyzed by the Shapiro–Wilk normality test. Statistical significance was determined with a two-tailed paired Student’s *t*-test comparison between two groups of data sets, and among three or more groups was analyzed by one-way analysis of variance (ANOVA). Differences were considered significant at * *p* < 0.05, ** *p* < 0.01. Statistical analyses were performed using SPSS for Windows (SPSS, Chicago, IL, USA).

## 3. Results

### 3.1. Structure–Activity Relationship (SAR)

In order to improve the anti-proliferative effect of tryptanthrin, twelve tryptanthrin derivatives were synthesized (Figure 1B). The inhibitory effect of these compounds (A1–A12) on the growth of Hep3B HCC cells was assessed after treatment for 48 h, using an MTT assay. Among these tryptanthrin derivatives, A6 exhibited the most potent inhibitory effect with an IC_50_ of 1.4 μmol/L (Figure 1B, doxorubicin as a positive control), relative to 5.7 μmol/L for tryptanthrin (Figure 2A). SAR analysis of the twelve compounds demonstrated that an electron-withdrawing group at C-8, such as bromine (e.g., A1) or iodine (e.g., A2), is essential for their anti-proliferative activity. In addition, compared with A4, which has a bromine group at C-2, A11 exhibited poorer cell growth inhibition in Hep3B cells. As shown in Figure 2B,C, A1 and A6 inhibited cell growth in a dose- and time-dependent manner in Hep3B cells. The IC_50_ values of sorafenib, a currently used agent in the treatment of HCC [25], were approximately 9.2 μmol/L and 4.8 μmol/L after 24 h and 48 h treatment in Hep3B cells (Figure 2D). Since the IC_50_ values of A1 and A6 were lower than those of the others, the pharmacological mechanism of these two compounds was investigated in subsequent experiments.

### 3.2. A1 and A6 Induce Cell Cycle Arrest

To examine the effect of the tryptanthrin derivatives on cell cycle progression in HCC cells, Hep3B cells were treated with A1 or A6 for 48 h and stained with propidium iodide (PI). Flow cytometric analysis demonstrated that A1 arrested Hep3B cells in the S phase in a dose-dependent manner (Figure 3A, etoposide as a positive control). After treatment with 5 μmol/L A1, the cell population in the G2/M phase increased from 11.9% to 50.8% in the control group (Figure 3A). Likewise, the proportion of cells in the G2/M phase also decreased in A6-treated Hep3B cells. The effect of A1 and A6 on the proteins involved in regulating the S and G2/M phases was also investigated. Western blotting showed that A1 and A6 downregulated the levels of cell cycle-related proteins, including cyclin A1, cyclin B1, CDK2, and p-CDC2 in Hep3B cells (Figure 3B).

### 3.3. A1 and A6 Induce Caspase-Dependent Apoptosis in Hep3B Cells

PI/Annexin V staining indicated that A1 and A6 increased the percentage of apoptotic cells in a dose-dependent manner after 48 h of treatment in Hep3B cells (Figure 4A). Compared with 4.9% in the control group, the percentage of double-stained cells considerably increased to 51.5% and 28.5% after treatment with 2.5 μmol/L A6 and 2.5 μmol/L A1, respectively (Figure 4A). Additionally, flow cytometric analysis showed that both A1 and A6 induced caspase-3 activation (Figure 4B, staurosporine as a positive control). Western blotting demonstrated that the level of caspase-8 was downregulated by both A1 and A6 in a dose-dependent manner in Hep3B cells (Figure 4C).

### 3.4. A1 and A6 Modulate Signaling of Akt and Mitogen-Activated Protein Kinases (MAPKs)

Multiple studies have shown that activation of Akt and MAPKs is involved in the occurrence and development of liver inflammation, accounting for a correlation between HCC aggressiveness and poor prognosis [26,27,28]. Western blotting showed that A1 and A6 induced downregulation of Akt phosphorylation and a downstream effector, NF-κB, in a dose-dependent manner with no obvious impact on the level of IκBα, the inhibitor of NF-κB, in Hep3B cells (Figure 5A). In addition, the level of NF-κB was decreased after treatment of A1 and A6 in the nucleus of Hep3B cells (Figure 5B). Furthermore, A1 and A6 decreased the levels of MAPK family signaling cascades, including p-ERK and p-JNK, accompanied by an increase in p-p38 (Figure 5A). A p38 inhibitor, SB203580, was used to examine the upregulation of p-p38 in A6-treated cells. Western blotting demonstrated that the level of p-p38 after treatment with A6 and SB203580 was lower than that in the group treated with only A6 (Figure 5C). The cell viability of the group treated with a combination of A6 and SB203580 was assessed using an MTT assay. Pre-treatment with SB203580 did not interfere with A6-induced cytotoxicity, suggesting that p38 might not be the major target of A6 in HCC cells (data not shown).

### 3.5. A1 and A6 Increase Generation of Reactive Oxygen Species

Previous studies have shown that the augmentation of oxidative stress contributes to the progression of HCC [29,30]. We found that A1 and A6 increased ROS generation in a concentration-dependent manner in Hep3B cells (Figure 6A, H_2_O_2_ as a positive control). As shown in Figure 6B, pre-treatment with the ROS scavenger *N*-acetylcysteine (NAC) or glutathione (GSH) partially reversed A6-induced ROS production. A similar phenomenon was observed in A1-treated Hep3B cells (Figure 6B). Western blotting showed that A1 and A6 upregulated the phosphorylation of p53, a DNA damage response biomarker [31], in Hep3B cells (Figure 6C). Furthermore, the level of p-p53 was also increased in another HCC cell line (SK-Hep1 cells) expressing wild-type p53 [32] after treatment with A1 and A6 (Figure 6D). To investigate the role of p53 in A1- and A6-treated cells, pifithrin-α, a p53 inhibitor, was used. As shown in Figure 6E, the level of p-p53 after the combination of A1 and pifithrin-α was lower than that in the A1 alone group. A similar phenomenon was observed in A6-treated Hep3B cells (Figure 6F). Compared with A1 or A6 alone, PI/annexin V double staining showed no significant change in the apoptotic cell count upon combined treatment of A1 or A6 and pifithrin-α (data not shown). These results suggested that p53 might not be the main molecule required for A1 or A6-induced apoptosis in HCC cells.

## 4. Discussion

It is well known that chronic inflammation caused by the hepatitis virus leads to the occurrence of HCC [33]. Tryptanthrin is an anti-viral and anti-infective compound present in Ban-Lan-Gen, which possesses preventive and therapeutic properties against viral infection and inflammation as a TCM [10,15]. Tryptanthrin has been reported to exert anti-proliferative effects on human skin cancer cells [19] and human neuroblastoma cells [34] as well as induction of apoptosis in human breast cancer cells via GSTpi and c-junk NH_2_-terminal kinase (JNK) [35]. In the current study, we reported the pharmacological investigation of the tryptanthrin analogs A1 and A6 in HCC cells after the structural optimization. 

The replacement of hydrogen on C-8 imparted A6 with a higher potency for inhibiting cell growth. The derivative A1 with an 8-bromo substituent had superior anti-proliferative activity than the derivative A11 with a 2-bromo substituent and the parent compound (tryptanthrin) with no substituents at C-2 and C-8. Compared with sorafenib, A1 and A6 demonstrated a higher anti-proliferative effect in Hep3B cells. Qin et al. reported that metal-tryptanthrin complexes cause S-phase arrest by inhibiting telomerase in bladder cancer cells [36]. Benzo[*b*]tryptanthrin suppresses cell growth by inhibiting Topo II, a ubiquitous enzyme that is essential in the regulation of DNA topology [37], in breast cancer cells [38]. In the present study, A1 caused S and G2/M arrest, while A6 affected cell arrest in the G2/M phase in Hep3B cells. Both A1 and A6 downregulated the levels of cyclin A1, cyclin B1, CDK2, and p-CDC2. Cyclin A1, a rate-limiting protein for the G1/S transition, interacts with CDK2 for the onset of both DNA replication and mitosis [39]. Additionally, the formation of the cyclin B1/CDC2 complex regulates entry into the M phase [40]. 

Apoptosis plays a pivotal role in the pathogenesis of many diseases, including cancer, thereby providing a strategic target in tumor therapy [41]. Yu et al. reported that tryptanthrin-derived indoloquinazolines induce apoptosis through caspase-3/7 activation in breast cancer cells [42]. Our results showed that A1 and A6 increased caspase-3 activity and decreased the expression of caspase 8. It is worth noting that the deregulation of Akt and MAPK pathways is implicated in HCC carcinogenesis [43,44]. In a clinical study, activation of the PI3K/Akt pathway was correlated with tumor progression and decreased survival rate in HCC patients [45]. NF-κB, a pleiotropic transcription factor, regulates inflammation and cell survival, playing an important role in inflammation and tumorigenesis [46]. Our results demonstrated that A1 and A6 downregulated the levels of p-Akt and p-NF-κB, accompanied by the inhibition of the translocation of NF-κB to nuclei in Hep3B cells. The level of IκBα was not changed which suggested that inactivation of NF-κB of A1 and A6-treated cells might be involved in the non-canonical NF-κB pathway [47]. It has been reported that p38 enhances the production of proinflammatory cytokines and controls the proliferation of lung progenitor cells and hepatocytes [43]. Yu et al. reported that tryptanthrin induces apoptosis by decreasing the interaction between GSTpi and JNK in doxorubicin-resistant breast cancer cells [35]. In the present study, we observed that the levels of p-JNK were diminished after treatment with A1 and A6. Shankar et al. reported that tryptanthrin suppressed phorbol 12-myristate 13-acetate (PMA)-induced p-p38 expression in skin cancer cells [19]. However, our results showed that A1 and A6 inhibited p-ERK and increased p-p38 in Hep3B cells. This discrepancy might be attributed to structural modifications and the use of different cancer cell lines. 

Previous studies have shown that strong oxidative stress, including exposure to hepatitis virus, chronic inflammation, and DNA damage, is associated with the progression of HCC [31,48,49]. Tsukiyama-Kohara et al. reported that gene expression of hepatitis C virus suppresses p53 activity which, in turn, causes apoptotic resistance to oxidative stress [50]. Interestingly, a recent study reported that the crude extract of Ban-Lan-Gen protected against hydrogen peroxide-induced injury in neuroblastoma cells [51]. Coriat et al. reported that sorafenib induced HCC cell death through increasing ROS production and the higher concentration of serum of advanced oxidation protein products is an early predictor in HCC patients which treated with sorafenib [52]. Similar to sorafenib, our results revealed that A1 and A6 induced ROS generation in Hep3B cells. Meanwhile, we found that A1 and A6 increased the phosphorylation of p53 in both Hep3B and SK-Hep1 cell lines. Additionally, we observed that A1- and A6-induced apoptosis was not affected by the administration of pifithrin-α. Taken together, the above results suggested that the efficacy of A1 and A6 against HCC cells might be impactful in the therapy of other solid tumors.

Although our results have demonstrated the anti-tumor effects of A1 and A6 in HCC cells, several limitations of our study should be noted. First, the reversal of A1- and A6-mediated cell death was not observed in the combination of SB203580 or pifithrin-α, which suggested that other molecules except p38 and p53 were involved. Second, although A1 or A6 treatment increased ROS generation, it is still unknown whether ROS exert their effects in A1- or A6-mediated apoptosis. In addition, the *in vivo* data of A1 and A6 has not been performed in this present study. 

## 5. Conclusions

In summary, our data suggest that tryptanthrin derivatives A1- and A6-induced cytotoxicity in HCC cells partly through caspase activation, ROS generation, and modulating Akt and MAPK signaling. Based on these findings, A6 is the most potent anti-tumor agent among all the synthesized tryptanthrin derivatives and might be considered a promising lead compound for the further development of therapeutic agents for HCC therapy.

## Figures and Tables

**Figure 1 biomedicines-09-01527-f001:**
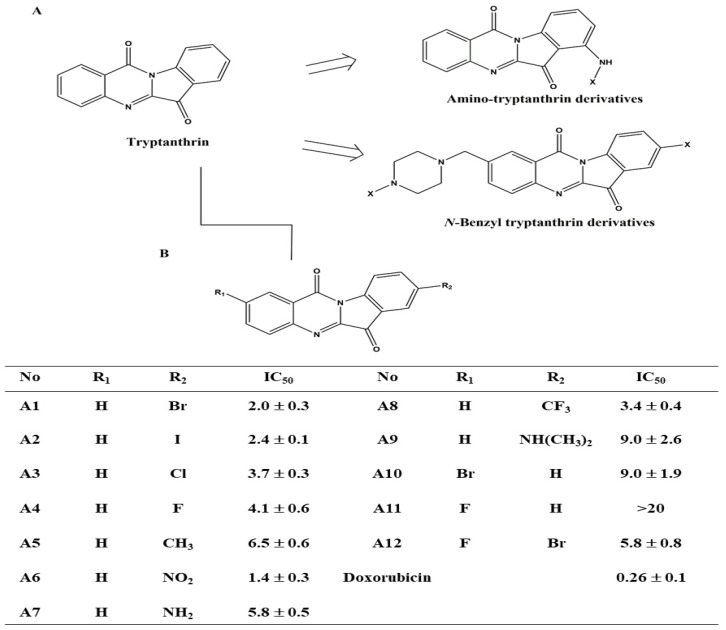
Tryptanthrin as the lead compound to synthesize anti-tumor agents. (**A**) Structure of tryptanthrin. (**B**) Structures and potencies of the tryptanthrin derivatives (A1–A12) in Hep3B cells. Cell viability was examined via MTT assay and the IC_50_ values were measured 50% relative to DMSO.

**Figure 2 biomedicines-09-01527-f002:**
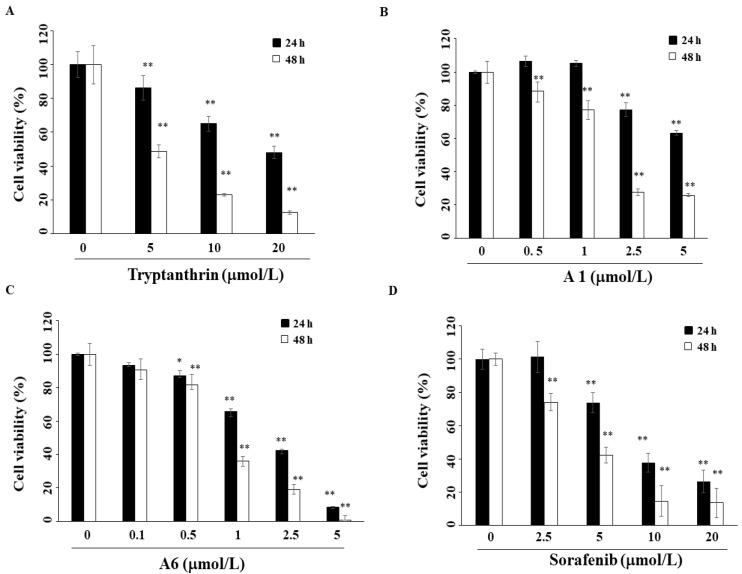
(**A**) Anti-proliferative effects of tryptanthrin, (**B**) A1, (**C**) A6, and (**D**) sorafenib in Hep3B cells. Cells were treated with the above compounds for 24 h or 48 h, and cell viability was detected via MTT assay. *Points*, means; *bars*, S.D. (*n* = 3). * *p* < 0.05, ** *p* < 0.01.

**Figure 3 biomedicines-09-01527-f003:**
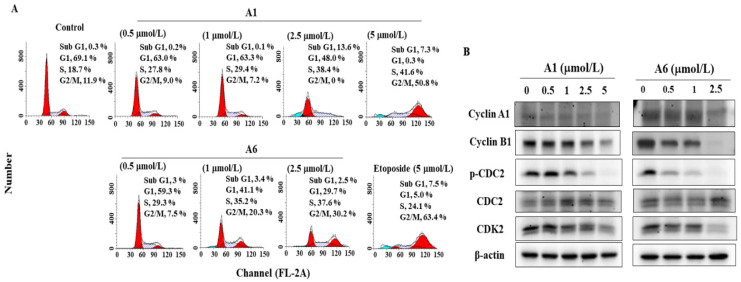
A1 and A6 induce cell cycle arrest. (**A**) Cell cycle analysis of A1- and A6-treated Hep3B cells for 48 h. Cell cycle population in each histogram was indicated. The first peak is the G1 phase and the second peak is the G2/M phase. Etoposide (5 μmol/L) was used as a positive control. (**B**) Effects of A1 and A6 on the phosphorylation/expression of cyclin A1, cyclin B1, CDC-2, and CDK2 in Hep3B cells.

**Figure 4 biomedicines-09-01527-f004:**
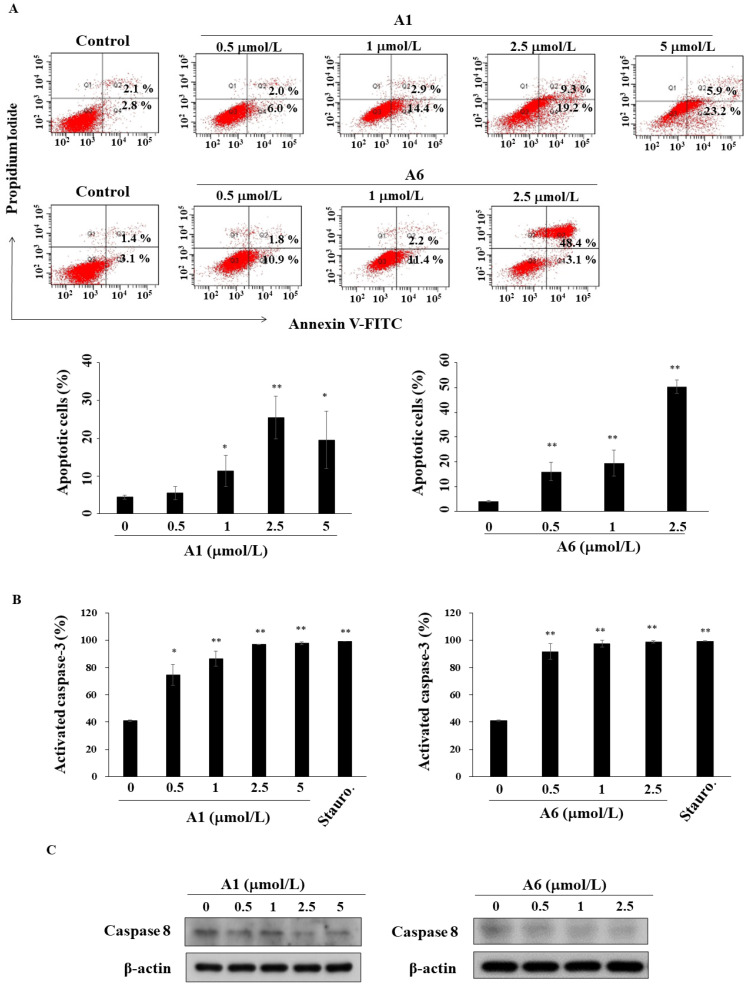
A1 and A6 induce apoptosis in Hep3B cells. (**A**) Upper panel, flow cytometric analysis of apoptotic cells after treatment with A1 or A6 for 48 h in Hep3B cells. Lower panel, statistical analysis of apoptotic cells after treatment with A1 or A6 in Hep3B cells. *Points*, means; *bars*, S.D. (*n* = 3). * *p* < 0.05, ** *p* < 0.01. (**B**) Caspase-3 activation by A1 and A6 in Hep3B cells using flow cytometry. Saturosporine (Stauro., 25 nmol/L) was used as a positive control. *Points*, means; *bars*, S.D. (*n* = 3). * *p* < 0.05, ** *p* < 0.01. (**C**) Effects on caspase-8 after treatment with A1 and A6.

**Figure 5 biomedicines-09-01527-f005:**
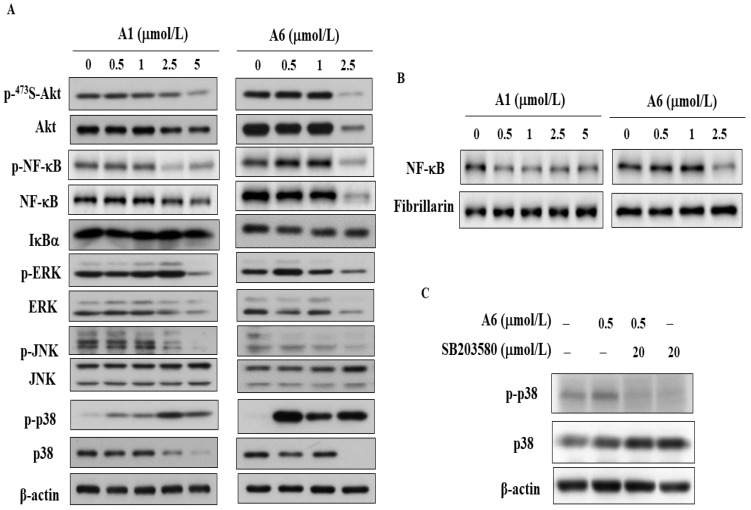
Effects of A1 and A6 on Akt/NF-κB and MAPKs in Hep3B cells. (**A**) Phosphorylation/expression of Akt, NF-κB, IκBα, ERK, JNK, and p38 after treatment of Hep3B cells with A1 or A6. (**B**) Western blot analysis of the nuclear expression of NF-κB. Proteins from nuclear cellular fractions were isolated from Hep3B cells treated with A1 or A6 for 48 h. Fibrillarin was used as a nucleus-specific loading control. (**C**) Phosphorylation and expression of p38 in cells treated with A6 (0.5 μmol/L) alone or pre-treated with SB203580 (20 μmol/L) for 15 min in Hep3B cells.

**Figure 6 biomedicines-09-01527-f006:**
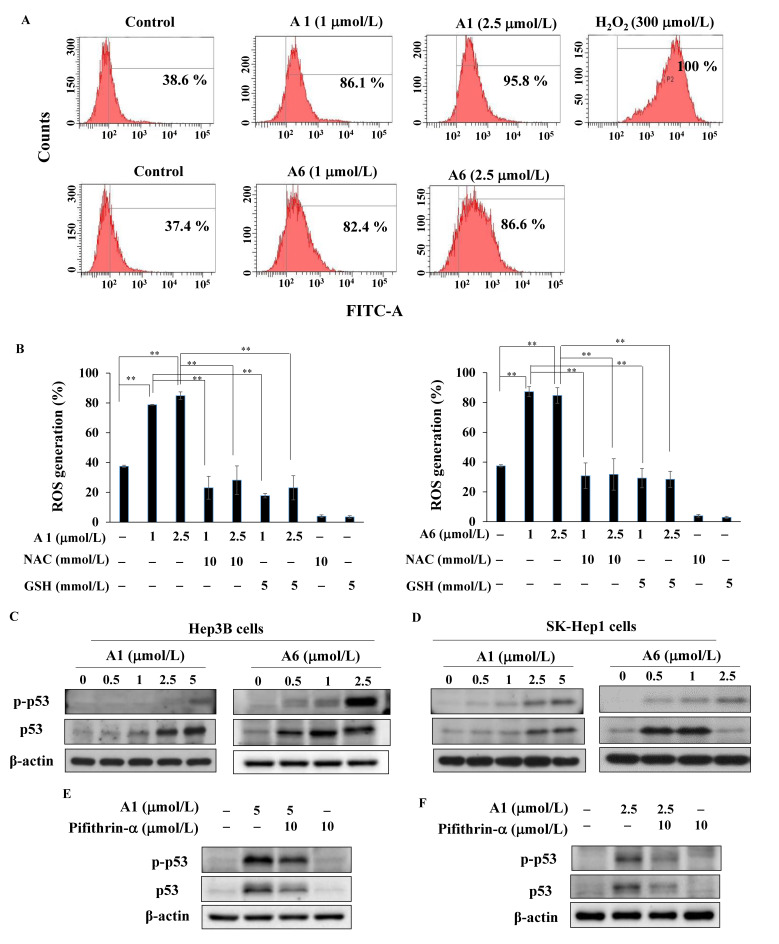
Analysis of reactive oxygen species (ROS) in A1- and A6-treated HCC cells. (**A**) Cells were treated with A1, A6, or DMSO for 3 h and stained with carboxy-DCFDA. H_2_O_2_ (300 μmol/L) was used as a positive control. (**B**) Left, cells were treated with A1 or in combination with *N*-acetylcysteine (NAC) or glutathione (GSH) for 3 h. Right, cells were treated with A6 or in combination with NAC or GSH for 3 h. Data are presented as the mean ± S.D. (*n* = 3). ** *p* < 0.01. (**C**) Expression of p-p53 and p53 after treatment with A1 and A6 in Hep3B cells and (**D**) SK-Hep1 cells. Phosphorylation and expression of p53 in Hep3B cells treated with (**E**) A1 (5 μmol/L) and (**F**) A6 (2.5 μmol/L) alone or co-treated with pifithrin-α (10 μmol/L) for 48 h.

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
