# Peer review of "Synthetic Tryptanthrin Derivatives Induce Cell Cycle Arrest and Apoptosis via Akt and MAPKs in Human Hepatocellular Carcinoma Cells"

_biomedicines, 2021, doi:10.3390/biomedicines9111527_

Round 1

Reviewer 1 Report

The authors have conducted appreciable work overall. Because the treatment options available for HCC are dire, this work holds importance to the current field. 

Results 

3.1 - To compare with the available and standard of care therapy, comparison of cell death induced by A1 and A6 to Sorafenib induced cell death is necessary. 

Figure 2 - Why is cell viability high at 48h and low at 24h? is 24h and 48h data flipped?

 Figure 3: Legend for the peaks is missing

Figure 4A: The data's compensation is not good. Fix the compensation.

Figure 5: Decrease in NFkB levels does not necessarily say that the activity is decreasing. There could be a parallel increase in S536P-P65, which suggests increase in activity. On the contrary, if the levels are IkB are increasing, it suggests that NFkB activity is decreasing. Hence, evaluate both S536P-P65 and IkB levels.

Results 3.5: ROS levels are increased on treatment with A1 and A6. However, A1 and A6 induce cell death and P-p53 levels in parallel. To offer explanation and to strengthen the claims, I suggest to inhibit the activity of P53 by inhibiting its phosphorylation + A1 and A6 drugs and evaluate the cell death.

All of the above data will loose credibility if the effectiveness of the drugs in controlling the tumor growth in-vivo is not evaluated. Hence, suggesting to perform xenograft tumor growth evaluation over a period of 30 days by including vehicle control, A1, A6, and sorafinib. 

For in-vivo data, suggesting to evaluate the weight of the nude mice during the treatment regimen.  

Author Response

Comment #1. To compare with the available and standard of care therapy, comparison of cell death induced by A1 and A6 to Sorafenib induced cell death is necessary.

Response:  Thank you for the comment. As suggested comments, the effects of sorafenib on cell viability has been added to compare A1 and A6 with sorafenib in the new Fig. 2 D.

Comment #2. Figure 2 - Why is cell viability high at 48h and low at 24h? is 24h and 48h data flipped?

Response: We are sorry for the mistaken. The cell viability of these compounds at 48 h are lower than 24 h. The marks of 24 h and 48 h have been revised in the new Fig. 2.

Comment #3. Figure 3: Legend for the peaks is missing

Response: Thank you for the comment. Per suggestion, the legend for the peaks have been added (Page 6, lines 165-166).

Comment #4. Figure 4A: The data's compensation is not good. Fix the compensation.

Response: Thank you for the comment. Per suggestion, the compensation have been revised in the new Fig. 4A.

Comment #5. Figure 5: Decrease in NFkB levels does not necessarily say that the activity is decreasing. There could be a parallel increase in S536P-P65, which suggests increase in activity. On the contrary, if the levels are IkB are increasing, it suggests that NFkB activity is decreasing. Hence, evaluate both S536P-P65 and IkB levels.

Response: Thank you for the suggestion. The levels of p-NF-kB and IkBa have been added in the new Fig. 5A. Also, we have provided the level of NF-κB was decreased after treatment of A1 and A6 in the nucleus of Hep3B cells in the new Fig. 5B (page 9, lines 191-193, page 10, lines 206-208).

Comment #6. Results 3.5: ROS levels are increased on treatment with A1 and A6. However, A1 and A6 induce cell death and P-p53 levels in parallel. To offer explanation and to strengthen the claims, I suggest to inhibit the activity of P53 by inhibiting its phosphorylation + A1 and A6 drugs and evaluate the cell death.

Response: The p53 inhibitor, pifithrin-a, has been co-treated with  A1 or A6, and PI/annexin V double staining showed no significant change in the apoptotic cell count upon combined treatment of A1 or A6 and pifithrin-a (data not shown). These results suggested that p53 might not be the main molecules required for A1 or A6-induced apoptosis in HCC cells. (page 11, lines 220-226).

Comment #7. All of the above data will loose credibility if the effectiveness of the drugs in controlling the tumor growth in-vivo is not evaluated. Hence, suggesting to perform xenograft tumor growth evaluation over a period of 30 days by including vehicle control, A1, A6, and sorafinib. 

Response: Our main goal for the present study is to provide in vitro efficacy and mechanism data for A1 and A6 in HCC cells in order to lay the groundwork and provide the rationale for its continued preclinical development in this regard. While the evaluation of A1 and A6 in animal models of HCC is a high priority for our group, in accordance with the three R’s for animal use (Replacement, Refinement, Reduction), we aim to have the present framework published while taking great care in selecting the model and study design that will most accurately predict the clinical success of A1 and A6. We look forward to comparing the in vivo results with the in vitro data in a future manuscript.

Comment #8. For in-vivo data, suggesting to evaluate the weight of the nude mice during the treatment regimen.  

Response: Thank you for the suggestion. The nude mice for evaluating the efficacy of A1 and A6 will be our first priority for our group in the near future.

Reviewer 2 Report

Thank you very much for allowing me to review the paper entitled " Synthetic tryptanthrin derivatives induce cell cycle arrest and apoptosis via Akt and MAPKs in human hepatocellular carcinoma cells" (Journal: Biomolecules 2021). The present study suggests the tryptanthrin analogues A1 (the derivative A1 with 8-bromo substituent) and A6 (the derivative A1 with 8-NO2 substituent) have potential clinical application since they target diverse aspects of cancer cell growth in human hepatocellular carcinoma.

Paper is based on the rich literature (49 items, of which 73% are from the last TEN years).

The names of the statistical tests used to check the distribution of the data (normality, non-normality) and the variance distribution are missing in MATERIALS AND METHODS.

In TEXT and Figures (2-6) should be introduced correct unit: “mol/L” instead “M”

In DISCUSSION:  lack of probable information about clinical consequences connected with the sentence “In contrast, our results revealed that A1 and A6 induced ROS generation, and this phenomenon could be reversed by the antioxidants NAC and GSH.

Lack of LIMITATIONS of study (at the end of DISCUSSION).

Author Response

Comment #1. The names of the statistical tests used to check the distribution of the data (normality, non-normality) and the variance distribution are missing in MATERIALS AND METHODS.

Response: Thank you for the suggestion. Per the reviewer’s suggestion, the data were analyzed by Shapiro-Wilk normality test, and the other information have been added in Materials and Methods section (Page 3, lines 116-122).

Comment #2: In TEXT and Figures (2-6) should be introduced correct unit: “mol/L” instead “M”

Response: It has been corrected per suggestion.

Comment # 3: In DISCUSSION:  lack of probable information about clinical consequences connected with the sentence “In contrast, our results revealed that A1 and A6 induced ROS generation, and this phenomenon could be reversed by the antioxidants NAC and GSH.

Response: Thank you for the suggestion. According to Coriat et al, sorafenib induced HCC cell death through increasing ROS production and the higher concentration of serum of advanced oxidation protein products is an early predictor in HCC patients which treated with sorafenib (Coriat et al, Mol Cancer Ther 2012;11:2284). Similar to sorafenib, our results revealed that A1 and A6 induced ROS generation in Hep3B cells (Page 14, lines 289-294).

Comment # 4: Lack of LIMITATIONS of study (at the end of DISCUSSION).

Response: The limitations have been added (Page 15, line 300-306).